# Efficacy of DNA Vaccines in Protecting Rainbow Trout against VHS and IHN under Intensive Farming Conditions

**DOI:** 10.3390/vaccines10122062

**Published:** 2022-12-01

**Authors:** Andrea Marsella, Francesco Pascoli, Tobia Pretto, Alessandra Buratin, Lorena Biasini, Miriam Abbadi, Luana Cortinovis, Paola Berto, Amedeo Manfrin, Marco Vanelli, Simona Perulli, Jesper S. Rasmussen, Dagoberto Sepúlveda, Niccolò Vendramin, Niels Lorenzen, Anna Toffan

**Affiliations:** 1Istituto Zooprofilattico Sperimentale delle Venezie, National Reference Laboratory for Fish Diseases, 35020 Legnaro, Italy; 2FATRO S.p.A., 40064 Ozzano dell’Emilia, Italy; 3Unit for Fish and Shellfish Diseases, Institute for Aquatic Resources, Technical University of Denmark, Kemitorvet, Building 202, DK-2800 Kgs. Lyngby, Denmark

**Keywords:** DNA vaccine, rainbow trout, VHS, IHN, field trial

## Abstract

Despite the negative impact of viral hemorrhagic septicemia (VHS) and infectious hematopoietic necrosis (IHN) on European rainbow trout farming, no vaccines are commercially available in Europe. DNA vaccines are protective under experimental conditions, but testing under intensive farming conditions remains uninvestigated. Two DNA vaccines encoding the glycoproteins (G) of recent Italian VHSV and IHNV isolates were developed and tested for potency and safety under experimental conditions. Subsequently, a field vaccination trial was initiated at a disease-free hatchery. The fish were injected intramuscularly with either the VHS DNA vaccine or with a mix of VHS and IHN DNA vaccines at a dose of 1 µg/vaccine/fish, or with PBS. At 60 days post-vaccination, fish were moved to a VHSV and IHNV infected facility. Mortality started 7 days later, initially due to VHS. After 3 months, IHN became the dominant cause of disease. Accordingly, both DNA vaccinated groups displayed lower losses compared to the PBS group during the first three months, while the VHS/IHN vaccinated group subsequently had the lowest mortality. A later outbreak of ERM caused equal disease in all groups. The trial confirmed the DNA vaccines to be safe and efficient in reducing the impact of VHS and IHN in farmed rainbow trout.

## 1. Introduction

Rainbow trout (*Oncorhynchus mykiss*) is the most farmed finfish in Italy, reaching 33,230 metric tons produced in 2020 (FAO, Fishstatj). Amongst the major threats to rainbow trout productions there are two viral diseases: viral hemorrhagic septicemia (VHS) and infectious hematopoietic necrosis (IHN), which are both caused by viruses belonging to the *Novirhabdovirus* genus within the *Rhabdoviridae* family. These diseases are in fact the main cause of production losses in rainbow trout farming [1] and are endemic in many European member states. Notably, both VHS and IHN are WOAH (former OIE) notifiable diseases and subjected to regulations in the European Union (EU). According to the Commission Implementing Regulation (EU) 2018/1882, VHS and IHN are classified as category C, D, and E diseases. Consequently, rules are in place in order to prevent their spread within the EU, in particular to member states that are officially free or have ongoing eradication programs. Although VHS is generally considered a more severe disease with more negative impacts on productions than IHN, a progressive increase of the IHN virulence has been observed in recent years [2]. Notably, IHN has been spreading in the last years in previously disease-free territories such as Finland in 2017 [3] and Denmark in 2021 [4]. In Italy, the majority of infected farms are located in northern regions, which are the main rainbow trout farming territories. In territories declared infected, vaccination may be implemented in synergy with stamping out, allowing reduction of disease prevalence prior to proceed with an eradication campaign. Currently, the new Animal Health Law, Regulation 2016/429 of the European Parliament, allows the member state to decide if and how it intends to use veterinary drugs for the control of the listed diseases with certain exceptions. However, there are no vaccines designed and licensed for use in rainbow trout against VHS or IHN in the EU. Traditional vaccines based on whole inactivated viruses can provide protective immunity, but are not cost efficient due to the antigen demand and needs for fish cell cultures for production [5]. Additionally, some inactivation protocols may cause denaturation of the viral surface glycoprotein G, the key protective antigen [6]. Importantly, DNA vaccines encoding the G protein have proved to be very efficient in protecting fish against VHS and IHN [7,8] and represent a cheaper alternative to other vaccine classes [9]. DNA vaccines typically consist of a plasmid vector containing an antimicrobial resistance gene, a bacterial origin of replication, a eukaryotic promoter (a *Cytomegalovirus*-derived promoter is usually used), the gene encoding the vaccine antigen, and a polyadenylation sequence terminating the transcription [10]. Once administered intramuscularly, DNA vaccines will drive synthesis of the viral G proteins in transfected cells (mainly myocytes) and elicit both humoral and cell-mediated immune responses [9,10]. The most efficient way to administer DNA vaccines to fish seems to be via intramuscular injection (IM). Oral delivery has been attempted but required excessive amounts of plasmid along with needs for encapsulation [11,12,13]. According to the few studies available, DNA vaccines proved to be safe in humans [14] and in fish [15], and represent a low environmental hazard [9]. Notably, according to the Cartagena Protocol (http://bch.cbd.int/protocol/text/ (accessed on 29 September 2022)) ratified by the European Union, naked DNAs and plasmids are not considered living modified organisms (LMO) and should not be considered genetically modified organisms (GMOs). Moreover, according to the Directive 2001/18/EC, animals injected with plasmids cannot be considered GMOs since the plasmid neither replicates autonomously nor integrate in the host genome, if not specifically designed to do so. This feature is extremely important for the authorization process of DNA vaccines. Until now, two DNA vaccines have been licensed for use in aquaculture: Apex-IHN^®^ (Elanco GmBH, Cuxhaven, Germany) against IHN and Clynav^®^ (Elanco GmBH, Cuxhaven, Germany) against pancreas disease. Both vaccines are licensed only for Atlantic salmon, *Salmo salar*. It is important to note that Clynav^®^ was the first DNA vaccine to be commercialized for use in veterinary animals in Europe and hereby represents a milestone in applied vaccinology. In light of the needs for vaccines against VHS and IHN in farmed RT in Europe, the aim of the present study was to prepare two DNA vaccines, one against VHS and one against IHN encoding the respective G proteins of recent Italian virus isolates, and to test them for efficacy and safety in rainbow trout under intensive farming conditions.

## 2. Materials and Methods

### 2.1. VHS and IHN Plasmids Expression in EPC Cell Line

The DNA vaccines were prepared by inserting RT-PCR amplicons of the viral G genes into the polylinker of the pVax1 vector (Invitrogen TM, Waltham, MA, USA). The VHSV DNA vaccine (pVax1-vhsG) G gene was derived from the VHSV/O.mykiss /I/TN/67/Feb15 strain isolated in Italy in 2015, while the IHN DNA vaccine (pVax1-ihnG) G gene was obtained from the IHNV/O.mykiss/I/TN/86/Feb15 strain isolated in Italy in 2015. Batches of plasmid DNA were purified from overnight cultures of transformed *E. coli* DH5a using anion exchange columns (HiSpeed Plasmid Giga EF Kit, Qiagen, Hilden, Germany) within a GMP-certified vaccine production facility. To evaluate the plasmid batches in terms of antigen expression, a cell transfection test was performed in *Epithelioma Papulosum Cyprini* (EPC) cells. In detail, EPC cells [16], maintained at 20 °C in minimum essential medium eagle (EMEM) (Merck KGaA, Darmstadt, Germany) with 1% commercial antibiotics mix (Merck KGaA, Darmstadt, Germany), L-glutamine, and fetal bovine serum (FBS) at 10% (*w*/*v*), were seeded on glass slides in 24-well plates. After 24 h, at approximately 80% cells confluency, the culture medium was removed from each well and replaced with the transfection solution. The latter included 250 µL of Optimem^®^ I-1x (GIBCO, ThermoFisher Scientific, Waltham, MA USA), 1 µg of plasmid, and 2 µL of JetPEI^®^ (Polyplus, Illkirch, France) pre-diluted in 50 µL of the supplied buffer. In each plate, the plasmids for VHS, IHN, and the combination of both were inoculated in duplicate. As negative controls, the wells were inoculated with the transfection solution without the plasmid. The plates were slowly stirred for 10 min and then incubated at 25 °C for 6 h. At the end of the transfection period, the solution was removed from the wells and replaced with 500 µL of 2% FBS EMEM. The incubation continued at 15 °C until the immunofluorescence (IF) test, which was performed on 5, 7, and 10 days post-transfection. Before proceeding with the IF, the plates were examined by light microscopy. The culture medium was removed from the plate and the cells fixed with 80% acetone–phosphate-buffered-saline (PBS) for 30 min at room temperature. After fixation, the acetone was removed and the plates were washed three times by stirring with 0.05% PBS-Tween for 5 min. A total of 200 µL of in-house produced rabbit anti-VHS and anti-IHN polyclonal sera commonly used for diagnostic purposes at IZSVe were inoculated in wells containing the relative plasmids; one of the wells was used for negative controls. Sera were left to react for 30 min at 37 °C. Subsequently the plates were washed three times as previously described. A total of 200 µL of FITC-conjugated goat anti-rabbit serum (Sigma Aldrich, St. Louis, MI, USA) was inoculated into all the wells and allowed to react for 30 min at 37 °C. The plates were washed again. The slides were removed from the wells and mounted on a drop of 50% glycerol in PBS. The slides were read with a Zeiss Axioskop fluorescence microscope (Zeiss, Jena, Germany) and with a Leica TCS SP8 confocal microscope (Leica, Wetzlar, Germany).

### 2.2. Fish

Fish health status was assessed before each experiment by parasitological, bacteriological examination, and by real-time PCRs targeting VHSV, IHNV, infectious pancreatic necrosis virus (IPNV), salmonid alphavirus (SAV), piscine orthoreovirus 3 (PRV-3), and *Renibacterium salmoninarum* the causative agent of the bacterial kidney disease (BKD) according to standard diagnostic protocols used at the IZSVe and DTU. Only negative fish batches were used for the experiments. For the trial performed under controlled laboratory conditions, rainbow trout juveniles weighing 0.9 g were obtained from a VHS and IHN-free hatchery and introduced into the IZSVe experimental aquarium, kept in quarantine, and acclimatized for at least 10 days as required by institutional and national regulations regarding experimental animal welfare. The fish were housed in 300 L flow-through conical fiberglass tanks equipped with lids. The water temperature was kept at 12 °C and the dissolved oxygen above 6 ppm. Feeding was performed manually using the broadcasting technique several times in a day. The daily feed quantity varied between 4 and 1.5% of the biomass, according to the fish size. For the field trial, fifteen thousand rainbow trout juveniles weighting 7 ± 2 g were provided by a VHS and IHN disease-free hatchery and divided into three experimental groups of five thousand (*n* = 5000) fish each: one unvaccinated control, one vaccinated with 1 μg/fish of pVax1-vhsG, and one vaccinated with 1 μg/fish of pVax1-vhsG and 1 μg/fish of pVax1-ihnG. When housed in the VHS and IHN-free hatchery, the fish were housed in raceways receiving water from a spring well that provided a constant water temperature of approximately 10–11 °C. Standard procedures were applied for housing and manipulating the experimental fish throughout the trial in order to apply the same farming conditions used for the other farmed fish. The food was administered manually and several times a day in order to ensure an equitable distribution to all the fish. The daily feed ratio was determined according to the instructions provided by the manufacturer of the feed used in the farm (Aller Aqua A/S Christiansfeld, Denmark). All procedures were carried out in compliance with the National and Institutional regulations on Animal Welfare currently in force. The Italian Ministry of Health authorized the experimental use of fish for this study with the authorization protocol n° 105/2019-PR.

### 2.3. Viruses and Phylogenetic Analysis

The VHSV and IHNV strains used for the experimental challenge had already been characterized during a pathogenicity test conducted within the framework of a past IZSVe project (ANIHWA-NOVIMARK project) and proved to be highly virulent (data not shown). We selected the most recent viral strains that also induced mortality higher than 60%. For VHS, the VHSV/O. mykiss/I/TN/92-1/Feb19 strain (VHS 92-1) which induced 96% mortality was chosen, while the IHNV/O. mykiss/I/TN/220-2/Mar18 (IHN 220-2) strain was selected because its mortality reached 95%. Both viruses were chosen also due to their site of isolation, corresponding to the farm where the field trial would be performed. EPC cells were used to produce and titrate viruses for challenge according to standard procedures [17] and titers were calculated by an indirect method according to the formula of Reed and Muench and expressed as TCID_50_/mL [18]. The complete G gene sequence of VHSV was generated for strains used for the plasmid production, for the potency tests, and for the one isolated from the field trial, while for IHNV, the sequence was generated only from the field trial isolated strain as the other sequences were already available. In Table 1, details on all the viral strains relevant for the study are given. Adopted protocols for RNA purification, RT-PCR, and sequencing were retrieved from [19] with slight modifications. The obtained consensus G gene sequences were aligned and compared to reference sequences retrieved from GenBank through MEGA 11 software [20]. For both the VHSV and IHNV datasets, phylogenetic trees were inferred using the maximum likelihood method (ML) available in the RaxML program version 2.0, incorporating the GTR model of nucleotide substitution with the CAT model of rate heterogeneity among sites [21,22]. Bootstrap resampling (1000 replications) assessed the robustness of individual nodes of the phylogenies. Phylogenetic trees were visualized using the FigTree v1.4 software (http://tree.bio.ed.ac.uk/software/figtree/ (accessed on 29 September 2022)). The complete G gene sequences generated in this study were deposited in GenBank and assigned the following accession numbers: OP292649 for IHNV/O.mykiss/I/TN/15-18/Feb21; OP292651, OP292652, and OP292650 for VHSV/O.mykiss/I/TN/67/Feb15, VHSV/O.mykiss/I/TN/92-1/Feb19, and VHSV/O.mykiss/I/TN/15-15/Feb21, respectively. 

### 2.4. Controlled Conditions Trial

Upon reaching a size of about 10 g, the animals were moved to 80 L flow-through conical fiberglass tanks and divided in the experimental groups as reported in Table 2.

For both the potency and the safety test, the sample size included 34 animals. Regarding the potency test, it was assumed that the cumulative mortality of the infected unvaccinated control group would be at least 60%, and that the cumulative mortality in the vaccinated group should be lower than 25% to obtain a relative percentage of survival (RPS) > 75%. Therefore, wishing to highlight a 35% decrease in mortality linked to vaccination and considering α = 0.05 and a test power 1 − β = 0.85, the sample size for two proportion tests necessary to highlight this difference resulted to be equal to 34 animals per group. For the safety test, the same group size was used. In the vaccinated and challenged tanks, as well as in the negative control tank, four samplings of five subjects each were planned with the purpose of evaluating the fish response to vaccination. Therefore, 54 animals were introduced into each potency tank and 34 into each safety tank. In total, 608 animals were used. To evaluate the safety of DNA vaccines, fish were inoculated with double of the maximum dose foreseen by the experimental design. The animals were removed from the tank and anesthetized by bath with tricaine methanesulfonate (Tricaine^®^, Pharmaq, Overhalla, Norway) at a dose of 100 ppm. Upon reaching an adequate level of anesthesia, the animals were inoculated intramuscularly into the left epaxial muscle at the level of the dorsal fin with 2 µg of plasmid suspended in 100 µL of sterile PBS, before being reintroduced into the original tank. The animals were monitored to assess proper recovery and monitored for 14 days for the appearance of any anomalies and signs of suffering, according to the *European Pharmacopoeia*. At the end of the monitoring period, the animals were euthanized by overdose of tricaine methanesulfonate. Muscle samples at the injection site were taken from five subjects and fixed in 10% neutral buffered formalin for histological evaluation. For the potency test, fish were vaccinated singularly with the vaccine for VHS and IHN, and with both plasmids at two different doses. The animals were removed from the tank and anesthetized by bath with tricaine methanesulfonate at a dose of 100 ppm. Upon reaching an adequate level of anesthesia, the animals were inoculated intramuscularly into the left epaxial musculature at the level of the dorsal fin with 0.1 and 1 µg of plasmid suspended in 100 µL of sterile PBS, before being reintroduced into the tank. A mock vaccinated group was set up by inoculating fish with 100 µL of sterile PBS. The fish were observed for 60 days post-vaccination (dpv). Five randomly selected subjects were collected from each vaccinated tank and from the negative control tank at 24 h post-vaccination (hpv), 48 hpv, and 30 dpv for histological analysis. The epaxial muscle tissue adjacent to the injection site was excised, placed in 10% buffered formalin, and stored at room temperature for 48 h before processing for histology. After 60 dpv, the animals were bath infected with a predicted dose of 10^4^ TCID_50_/mL of the selected strains of VHSV and IHNV individually (tanks 7-8-9-10 immunized for single disease), or in combination (tanks 11-12 vaccinated against VHS and IHN). The respective unvaccinated control groups (tanks 4-5-6) were infected in the same way. Tank 3, the negative control group, was mock infected with sterile MEM. In detail, the water flow was stopped in each tank and the volume decreased to 20 L. Each viral solution was brought to a final volume of 200 mL by dilution in MEM in order to uniform the volumes used for the infection before being poured into the tanks. After 10 min, a water sample was collected from each tank to perform back-titration in cell culture according to the method previously described for viral titration. After 2 h, the flow and the volume of water were restored. The infected animals were observed daily for 30 days to monitor the progress of disease. The appearance of lethargy, irregular swimming, melanosis, and exophthalmos that generally precedes the death of the animals in natural infections was considered the humane endpoint of the study. All subjects showing the detailed clinical signs were euthanized by anesthetic overdose. Spleen and kidney were sampled from all the dead fish in the tanks treated with vaccines for both diseases and submitted to RNA extraction with QIAsymphony DSP Virus/Pathogen Midi kit (Qiagen, Hilden, Germany), in combination with the automated system QIAsymphony SP (Qiagen, Hilden, Germany). Real-time RT-PCR for VHSV and IHNV was subsequently performed according to the protocols described by [23,24] for IHNV and by [25] for VHSV in order to determine the virus responsible for the mortality. At the end of the observation period, the surviving animals were euthanized as described before. The muscle at the injection site was sampled from fish treated with the highest dose of vaccine to evaluate the plasmid persistence by end-point PCR (see dedicated paragraph).

### 2.5. Histological Analysis 

Muscle samples fixed in 10% neutral buffered formalin were processed for histopathological analysis according to standard procedures. Briefly, samples were dehydrated through a graded ethanol–xylene series and embedded in paraffin. Sections of 3 µm were first deparaffinized, rehydrated, and then stained with hematoxylin–eosin (H&E) for morphological examination. IHC was performed automatically using the BenchMark Ultra (Ventana Medical Systems inc. Instrument, Tucson, AZ, USA). Between each reagent application, the slides were washed with Tris-buffered saline with Tween 20 (Dako Denmark A/S, Glostrup, Denmark). Antigen retrieval was achieved by incubation at 36 °C for 32 min with protease 2 (Roche Diagnostics, Basel, Switzerland). Non-specific antibody binding sites were blocked with Discovery Goat Ig Block solution (Roche Diagnostics, Basel, Switzerland), by incubation at 36 °C for 28 min for VHSV and 4 min for IHNV. IHC reactions were performed with rabbit anti-VHS and anti-IHN polyclonal sera produced in-house commonly used for diagnostic purposes at IZSVe at the dilution of 1:10,000 and 1:500, respectively, for 1 h at room temperature. Samples were incubated with the secondary antibody conjugated with alkaline phosphatase UltraMap Rabbit AP for 16 min and then chromogen discovery red (Ventana Medical Systems Inc., Tucson, AZ, USA) was applied for 16 min. Slides were then counterstained with Mayer’s hematoxylin for 8 min, air-dried, dipped in xylene, and mounted in Eukitt medium (Kaltek, Saonara, Italy). Bright red IHC staining would highlight the presence of VHSV or IHNV G glycoproteins. Positive controls for VHS and IHN IHC methods were obtained via bath challenged juvenile specimens of *O. mykiss* (ANIHWA-NOVIMARK project). Moribund fish were fixed and processed whole for routine H&E morphological staining and IHC staining.

### 2.6. Field Trial

Based on the results obtained during the controlled conditions trial, the treatments at the dose of 1 μg/fish of VHS plasmid and the combination of both VHS and IHN plasmids were selected for the field trial. Vaccination procedures were planned for the beginning of October 2020, guarantying the availability of juveniles of the suitable size and allowing the introduction of the experimental fish into the infected grow-out at a water temperature ideal for VHS and IHN outbreaks to occur. Since 2015, the farm selected for the field trial has been undergoing yearly monitoring for VHSV and IHNV detection according to the surveillance plan adopted by the Trento Province where the farm is located. According to surveillance, the farm turned out to be infected with both viruses. Fasting was performed for 1 day before vaccination. Fish were anesthetized with tricaine methanesulfonate at a dose of 100 ppm. Upon reaching a satisfying level of anesthesia, fish were then moved to the working table, hosted in a semi-closed, easy-to-clean container. Here, they were injected intramuscularly into the epaxial musculature at the level of the dorsal fin with 0.05 mL of vaccine solution. The mock vaccinated group was mock injected with 0.05 mL of sterile PBS. Four Socorex^®^ ultra 1810 micro-range tube feeding syringes equipped with stainless steel 25 g × 3 mm needles were used for the injection. Once vaccinated, fish were transferred directly into their respective raceway. The mock vaccinated group was treated first, followed by the VHS vaccinated group, and the double vaccinated group being last. All procedures were performed in one day. At the end of the vaccination, all working areas and materials were sanitized with 1% *w*/*v* hydrochloric acid solution. Fish were checked for recovery and feeding was resumed 24 h after the vaccination. Mortality was checked daily and recorded. After 57 dpv, corresponding to approximately 630 degree/days, fish were weighted and moved to the infected grow-out. The fish were housed in a single raceway which was divided into three sections by using two metal grids. According to the water flow direction, the mock vaccinated group was introduced first followed downstream by the VHS vaccinated group and the bivalently vaccinated one as last. Water temperature was similar to the one present in the hatchery for all the period of observation ranging from 11 ± 1. The water supply of the raceway housing vaccinated fish comes from the disease-free hatchery where the fish were housed after vaccination; the water passes into the stocking channels for the slaughterhouse (see Appendix A). This feature facilitated the viral spread to the experimental fish. Daily mortality was recorded. At the onset of mortality and onwards, anatomopathological examinations were performed in order to identify the pathogens involved. Due to the high number of fish dying, the mock vaccinated group was checked weekly while fish from the vaccinated groups, which showed a more affordable level of mortality, were all collected and sampled daily. Each head kidney was sampled and tested by real-time RT-PCR for VHSV and IHNV. At 134 dpv, fish were counted to obtain more accurate mortality data. After this time, the daily examination of the affected fish was interrupted. Mortality was still recorded daily and diagnostic examinations were performed monthly in all the three groups until 270 dpv when the study ended.

### 2.7. Plasmid Detection

The persistence of the pVax1-vhsG and pVax1-ihnG plasmids in the vaccinated fish was assessed by semi-quantitative end-point PCR. For this purpose, samples of inoculated epaxial muscles were collected and preserved a −20°C at the end of the potency test (90 dpv). During the field trial, a muscle sample (1 cm × 1 cm) from the injection site was collected from five fish vaccinated with both plasmids at 130, 160, 180, 210, 230, 260, and 280 dpv for plasmid detection. At 320 dpv, muscle samples were obtained from fifteen fish from the same group. Total DNA was extracted using QIAamp^®^ Mini kit (Qiagen, Hilden, Germany) starting from 50 mg of muscle tissue and following the manufacturer’s instruction. DNA concentration was measured using the Qubit™ 1XdsDNA HS Assay Kit with the Qubit™ 4 Fluorometer (ThermoFisher Scientific, Waltham, MA, USA). Two distinct PCRs were designed to amplify pVax1-vhsG and pVax1-ihnG fragments. In addition, a fragment of the rainbow trout beta-globin gene was amplified under the same conditions, in order to evaluate the integrity and amplifiability of the extracted DNA. The primers set for the endogenous gene targeted two different exons of the *Oncorhynchus mykiss* beta-globin gene (LOC100136291). Primer sets were designed using the Primer3web version 4.1.0 (https://primer3.ut.ee/ (accessed on 29 September 2022)) and are specified in Table 3. PCR amplifications were performed with the Platinum™ Taq DNA Polymerase kit in a total volume of 25 μL containing: buffer 1X, MgCl_2_ 2.5 mM (1.5 mM for pVax1-vhsG PCR), 0.25 mM of dNTPs Mix, 0.5 mM of each primer, 1.25 U of Taq DNA polymerase, and 200 ng of sample DNA. The amplifications were performed as follows: initial denaturation for 2 min at 94 °C, 40 cycles of denaturation at 94 °C for 30 s, annealing at appropriate temperature (Table 3) for 30 s, and extension for 30 s at 72 °C, followed by a final extension step of 5 min at 72 °C. The amplified products were detected by electrophoresis on 1.5% *w*/*v* agarose gel stained with GelRed. Sensitivities of both plasmid detection PCRs were evaluated on known quantities of serially diluted plasmid (1:10) in rainbow trout genomic DNA matrix. Limit of detection (LoD) was 10^−7^ ng/reaction corresponding to about 20 plasmid copies for both pVax1-vhsG PCR and pVax1-ihnG PCR. Three positive controls containing progressively decreasing amounts of plasmid target (2 × 10^5^; 2 × 10^3^; 2 × 10^1^ copies) were included in each PCR reaction.

### 2.8. Statistical Analysis

For the analysis of the results obtained in the trial carried out under controlled conditions, the final cumulative mortality was calculated for each tank. The relative percentage of survival (RPS) was also calculated according to the following formula:(1)$$\mathrm{RPS}=\left[\;1-\frac{number\;of\;dead\;fish\;in\;vaccinated\;tank}{number\;of\;dead\;fish\;in\;untreated\;tank}\right]\times 100$$

The Kaplan–Meier analysis was used to analyze the effect of vaccination and for the graphic representation of the probability of survival during the challenge test. Once the proportionality of the risks was verified, the Cox semi-parametric model was used to compare the different experimental groups. To evaluate the equality of the survival functions of the different treatments, the log-rank test was used. Statistical analyzes were carried out using the STATA 12 statistical software. For the field trial mortality analysis, the same tests were applied to data related to 30 days post-transfer (dpt) and 60 dpt time-lapse. Moreover, due to an increase of mortality in all the experimental groups, including the bivalent vaccinated group, the same statistical tests were applied to mortality data observed between 200 and 240 dpv. 

## 3. Results

### 3.1. VHS and IHN Plasmids Expression in EPC Cell Line

Capability of the plasmids to drive expression of the viral G proteins was tested by immunofluorescence staining of transfected EPC cell cultures. Both plasmids proved to induce expression of the G glycoprotein on EPC cells, with first detection occurring from day five of incubation for the VHS plasmid. Some immunofluorescence positive reactions were observed at five days of incubation also for the IHN plasmid, but a clear positive signal was detected from day seven. The combination of both was tested singularly for VHS and IHN, and the same results were obtained (see Figure 1).

### 3.2. Viruses and Phylogenetic Analysis

All VHSV strains relevant for the study appeared to belong to the genetic cluster D within the sub-lineage Ia2 according to the classification by [19,26], together with some recent VHSV strains isolated in the Trento Province (Figure 2). Regarding IHNV, all the relevant strains belong to the sub-lineage E according to the classification proposed by [2,27]. Only the IHN 220-2, the strain used for the potency test, appeared to belong to a defined genetic cluster, namely cluster A, within sub-lineage E (Figure 3).

### 3.3. Controlled Conditions Trial

Following the vaccination procedures and during the bath infection, no signs of disease nor mortality appeared in any tank. No mortality occurred at any time during the trial in the negative control tank. During the safety test, fish did not show any behavioral alterations, lesions at the injection site, or any other signs of suffering. The results of the virus titrations carried out on the infection water samples are shown in Table 4. Viral titers were as expected in all the tanks, except for the co-infection tanks where the titer was higher, probably due to a synergic interaction between the two viruses. Within four days from the challenge, mortality appeared in all the tanks infected with VHSV, while it started later (from the eighth day onward) in those infected with IHNV only. Affected fish showed varying degree of skin melanosis, exophthalmos, anorexia, lethargy, and gill hemorrhages or anemia. Mortalities in the positive control groups were high in the tank infected with VHSV or with both viruses simultaneously (83.6% and 100%, respectively), while the tank infected with IHNV only had a mortality of 40.4%. The vaccine dose of 0.1 μg/fish resulted in a RPS of 49.2% against VHS, 44.6% against IHN, and 87.5% against co-infection. Higher RPS values were found for vaccine doses of 1 μg/fish namely: 78.5% against VHS, 72.5% against IHN, and 90% against the coinfection. The results are displayed in Table 4, while the outcome of the diagnostic real-time RT-PCR performed on the spleen and kidney samples taken from the dead/sick fish in tank 11 and 12, vaccinated with both plasmids and co-infected with both viruses are given in Table 5.

The survival curves obtained with the Kaplan–Meier analysis are shown in Figure 4. For the three groups analyzed (infected with VHSV, IHNV, and VHSV + IHNV), the hypothesis of proportionality of the risks was accepted, allowing the Cox method to be used for the comparison between the two vaccine doses (0.1 μg/fish versus 1 μg/fish). Cox's analysis showed a significant difference in the risk of event (death) between the treatments, with the exception of the bivalent formulation where no difference was found between the tested doses. Furthermore, the hypothesis of equality of the survival functions was rejected for all groups, supporting the statistical significance of the data obtained (*p* < 0.01).

### 3.4. Histology

In all experimental groups, histological evaluation performed on muscle samples taken at 24 h and 48 h after vaccination showed the presence of necrotic lesions associated with injection damage consisting of focal muscle fiber fragmentation and mild mononuclear inflammatory infiltration (Appendix A). Samples taken at the end of the safety test showed, for both plasmids, focal mononuclear inflammatory infiltrate around intact muscle fibers near the site of inoculation. Samples taken at 30 days after vaccination showed still the presence of varying degrees of mononuclear inflammatory infiltrate (see Appendix A), whereas mock vaccinated samples did not show any infiltrate in the musculature. IHC analysis did not give detectable staining in the musculature of all vaccinated groups (see Appendix A). 

### 3.5. Field Trial

In the five days following vaccination, approximately 0.5% mortality occurred in all groups and was attributed to manipulation. At 60 dpv, when fish were transferred in the infected facility and counted, the cumulative mortalities were 1.26, 1.22, and 2.32% in the mock vaccinated group, the VHS vaccinated group, and the VHS+IHN vaccinated group, respectively. At this time, the mean weight was 33 g in all groups. At 7 dpt, the mortality began simultaneously in the three experimental groups. The graph of daily mortality is shown in Appendix A. The mock vaccinated group showed the highest mortality rate with a peak on 11 dpt and, according to routine farming protocols adopted by the farm, this group was subjected to a 10-day fasting period. In the VHS vaccinated group, the mortality was less severe, reaching a peak approximately on 24 dpt. The VHS+IHN vaccinated group showed lower daily mortality compared to the other two groups without significant elevations. Results obtained by all diagnostic examinations are given in Table 6 and Table 7, while results of real-time RT-PCR for VHSV and IHNV performed daily on all moribund/dead fish collected from the VHS vaccinated group and the VHS/IHN vaccinated group are shown in Figure 5.

Initially, VHSV was identified as the main pathogen and therefore considered responsible for mortality in all the groups up to 30 dpt. However, IHNV subsequently emerged and progressively replaced VHSV as the cause of disease/mortality in the mock vaccinated group and in the VHS vaccinated group. In the VHS+IHN vaccinated group no obvious cause of mortality was detected and therefore the low mortality observed was attributed to the intensive farming conditions resulting in a certain background mortality. Due to the biphasic trend in the prevalence of the viral diseases and given the different types of vaccines used (monovalent vs bivalent), cumulative mortality and RPS data from vaccinated groups were evaluated at 30 and 60 dpt separately. Cumulative mortality data and RPS values are shown in Figure 6. At 30 dpt, cumulative mortality was 10.6%, 3.3%, and 1.0 % in the mock vaccinated group, the VHS vaccinated group, and the VHS+IHN vaccinated group, respectively. Based on these values, the vaccine-induced RPS for VHS was 69.3%, whereas the one of the bivalent formulation reached 90.5%. At 60 dpt, cumulative mortality increased to 18.4% in the mock vaccinated group, 7.6% in the VHS vaccinated group, and 2.5% in the VHS+IHN vaccinated group. The calculated RPSs for the two treatments were 59.6% for the VHS-only vaccine and 87.1% for the bivalent vaccine. The diagnostic examinations performed from 60 dpt onward generally tested positive for IHNV in all groups, while VHSV was detected at a decreasing frequency. In the period spanning from the end of April to the first half of May 2021, an unexpected increase in mortality in all three groups, including the VHS+IHN vaccinated group, was observed. Bacteriological tests turned out to be positive in all three groups for *Yersinia ruckeri* biotype II. Based on the antibiogram, a treatment with a trimethoprim-boosted sulfonamide administered per os for 10 days was implemented and resolved the bacterial disease. At the end of the observation period at 270 dpv, cumulative mortality was 35.3% in the control group, 26.0% in the VHS-vaccinated group, and 20.0% in the bivalent-vaccinated group; the mean weight was approximately 430.7 ± 143.7 g, 404.1 ± 106.4 g, and 429.9 ± 110.5 g in the mock vaccinated, the VHS vaccinated group, and the VHS+IHN vaccinated group, respectively.

Differences between mortality data of control groups and vaccinated groups obtained during the controlled settings trial proved to be statistically significant (*p* < 0.01). For the field trial, at 30 and 60 dpt, cumulative mortality appeared to be statistically different among all the experimental groups (*p* < 0.001 between mock vaccinated group and vaccinated groups and *p* < 0.001 between VHS vaccinated group and VHS+IHN vaccinated group). Analyses performed on data collected between 200 and 240 dpv showed no differences in mortality occurring in the experimental groups during this time-lapse (*p* > 0.01).

### 3.6. Plasmid Detection

During the trial under controlled conditions, the muscle samples collected from surviving fish in tanks 8, 10, and 12 at the end of the trial (90 dpv) were all positive (4/5) or weakly positive (1/5) for the presence of both plasmids. Results of plasmids detection in muscle tissue samples collected during the field trial are reported in Table 8. Both plasmids were detected with varying degrees of positivity in a portion of tested animals from 120 dpv until 260 dpv. At 280 dpv, all samples were negative. At 320 dpv, a larger set of samples were collected and six samples out of fifteen turned out to be weakly or very weakly positive for both plasmids. In addition, at 160 dpv, the presence of plasmids was also investigated in head kidneys and spleens and all samples were negative.

## 4. Discussion

Despite the eradication efforts, VHS and IHN re-occur periodically and cause significant economic damage to the Italian rainbow trout farming. Although vaccination could be a useful tool to mitigate the effects of these diseases, no licensed vaccines are currently available in Europe. However, DNA vaccination against IHN in Atlantic salmon has been used commercially in Canada since 2007 and VHS/IHN DNA vaccines have been shown to provide high protection both individually and in combination against the respective diseases in rainbow trout under experimental conditions [9,28]. This study therefore aimed at testing the potential of DNA vaccination for reducing losses due to IHN and VHS in rainbow trout under intensive farming conditions. The results confirmed the high efficacy and safety of the DNA vaccines developed and tested against recent Italian VHSV and IHNV strains, under both controlled and field settings. Notably, the field trial represents the first demonstration of the capacity of this vaccine technology to simultaneously protect rainbow trout against both diseases under farming conditions. The DNA vaccines were confirmed to be safe for the injected fish since no clinical side effects were observed and histological examinations of the injection site did not reveal any of the morphological lesions usually linked to vaccination such as granulomas or muscular fibrosis. The mononuclear inflammatory infiltrate detected up to 30 dpv probably was linked to expression of the viral G protein by transfected muscle cells as earlier reported [29], although this could not be confirmed by IHC, presumably due to the low vaccine dose. We here made an effort to design DNA vaccines encoding the G protein genes of recent local virus strains to reduce chances of reduced protection due to inter-strain variability. The evolution of IHNV has been reported to be more rapid than for VHSV [9], but it remains to be examined whether the resulting strain variability necessitates adaptation of the DNA vaccine. In the vaccination trial performed under controlled conditions, the mortality obtained in the IHN unvaccinated group was not as high as expected. A group of fish from the same batch were previously infected with the same IHN strain, IHN/O.mykiss/I/TN/220-2/Mar18, and in that case the mortality reached 95% (data not shown). However, in the present study the observed cumulative mortality reached only 38.18%. Thus, despite the statistically significant difference between the mortality presented by the unvaccinated and the vaccinated group, cumulative mortality was less than 60%. This is the value recommended by the *European Pharmacopoeia* for the evaluation of the efficacy of vaccination; therefore, the RPS data for this vaccine was not sufficiently robust to allow inclusion in the field trial. Consequently, only the monovalent formulation against VHS and the bivalent formulation against both viral diseases at a dose of 1µg/fish were selected for the field trial. In the infected facility, experimental groups were positioned in series within one raceway. After leaving the hatchery, the water supply was passing through the channel used for the housing of trout destined to be slaughtered. Consequently, the temperature of the water was comparable to that in the hatchery and constantly equal to 11 ± 2 °C. The probability of infection transmission was the highest due to the upstream position of older fish coming from the infected on-growing raceways, which instead received their water from the Sarca river. Furthermore, the serial position of the groups, with the control group upstream of the vaccinated ones, theoretically increased the viral load in the water. Due to the high number of animals involved in the field trial, it was not possible to test moribund fish of the mock vaccinated group individually, which instead was possible for the two vaccinated groups. Consequently, it was not possible to obtain exact data regarding the real impact of the two viral diseases on the control group. The comparison between unvaccinated and vaccinated groups was evaluated on the basis of anatomopathological examinations and real-time RT-PCR results obtained from vaccinated groups that allowed the cause of mortality to be divided into three phases: (i) VHS onset (up to 30 dpt); (ii) IHN onset (between 30 and 60 dpt); (iii) outbreak of yersiniosis (from 200 dpv to 240 dpv). As shown by statistical analysis, the mortality curves of the three groups were significantly different in the first two time-lapses (*p*= 0.0000) but were equal during the yersiniosis outbreak (*p*= 0.690). Since only a few dead fish were positive for IHNV (and none for VHSV), the increase in mortality in the VHS+IHN vaccinated group, observed during the bacterial disease outbreak, was not attributed to a decrease in the protection provided by the vaccines used in this group. The calculation of cumulative mortality and RPS provided by the vaccines showed that bivalent vaccination was effective in reducing the impact of VHS and IHN even under field conditions. Since the VHS+IHN vaccinated group assumingly underwent a more aggressive challenge than the other two groups due to its downstream positioning, the protection provided by the bivalent formulation appeared extremely robust both statistically and biologically. The evaluation of the efficacy of the vaccines tested under field conditions ended at 270 dpv (July 2021) and consequently it was not possible to monitor the persistence of the protection induced by the vaccines in the transitional period of autumn which, according to the farmers, represents a time of recrudescence of VHS and IHN. However, the animals had already reached a size compatible with marketing (430.7 ± 143.7 g, 404.1 ± 106.4 g, and 429.9 ± 110.5 g in the mock vaccinated group, VHS, and VHS+IHN vaccinated groups, respectively); therefore, the data collected were sufficiently representative of the protection provided by the vaccines during a normal production cycle for farmed rainbow trout in Italy. Regarding the VHS vaccine, the protection obtained was less effective, but this was due to the impact of IHN, a disease for which the animals were not vaccinated. It is well known that DNA vaccines induce a strong innate immune response in vaccinated fish [30]. This first response can provide cross-protection against viral diseases [28,31,32]. However, this first non-specific phase is not long lasting and is subsequently replaced by the specific immune response. The duration of each immune phase is temperature dependent. Indeed, cross-protection against VHS and IHN using the heterologous vaccine has been observed up to 40 dpv in fish acclimated at 5 °C and 10 °C, but not in fish acclimated at 15 °C [33]. Another study reported cross-protection lasting up to 880 degree/days at 10 °C [28]. In the present study, IHN impact showed to be more important in the group vaccinated against VHS compared to the group vaccinated against both diseases. This suggests that low or no level of cross-protection was present when the disease appeared in the experimental groups at approximately 650 degree/days. 

This study shows that DNA vaccines can effectively reduce the losses due to VHS and IHN also under intensive farming conditions. Extended testing with parallel positioning of experimental groups in replicate raceways/units will be useful for confirming the observations in a larger scale. This setup would require a full fish cohort to be vaccinated but would allow farming conditions to be even more comparable to those to which fish are subjected under intensive farming conditions. Prevalence reduction of VHS and IHN in the farm or in the territory subjected to vaccination in case more facilities are involved would be a useful feature to be investigated. However, further studies are needed to fully determine the real potential of the DNA vaccines as a tool for eradicating the diseases. In fact, DNA vaccines have proved to be effective in protecting fish against the disease but not from the infection. Vaccinated fish can get sub-clinically infected and represent a risk for naïve fish [34]. Results obtained in this study show that vaccinated fish surviving the first impact of IHN but constantly exposed to the virus can develop the disease subsequently. A study related to the efficacy of the Apex-IHN^®^ has shown that 100% vaccination of the fish population is required to prevent the spread of IHN. Conversely, vaccinated fish showed a reduced prevalence of the infection compared to unvaccinated fish [35]. Regarding the persistence of DNA vaccines inoculated into muscle in rainbow trout, our results show that they persist in this species for longer than what is currently known. The literature consulted and available in this regard reports a persistence of plasmid at least up to 45 days in rainbow trout if inoculated at a dose of 30 µg in adult subjects [36]. Note that in this study, the analyses were conducted up to 45 days after inoculation and were not prolonged. Analyses carried out on samples taken during the field trial showed the presence of plasmid inoculated at a dose of 1 µg /fish up to 320 dpv (corresponding to approximately 3520 degrees/day). This is similar to what has been reported for Atlantic salmon, in which a lucipherase-expressing plasmid inoculated at a dose of 100 µg/fish was found up to 535 dpv [37]. Although the persistence of the plasmid does not represent an obstacle to the marketing authorization of the vaccine, the results obtained warrant further investigations of the possible correlation between the duration of protection and the persistence of plasmids in the muscle of vaccinated fish.

## 5. Conclusions

The tested DNA vaccines have been found to be safe and efficient in reducing the impact of VHS and IHN in rainbow trout farmed under intensive conditions. In particular, the bivalent formulation is expected to represent a valuable tool for disease prophylaxis in farms encountering both diseases. Implementation may be expected to boost the rainbow trout production in endemically infected regions and at the same time improve the health status of the fish in VHS and IHN-infected farms.

## Figures and Tables

**Figure 1 vaccines-10-02062-f001:**
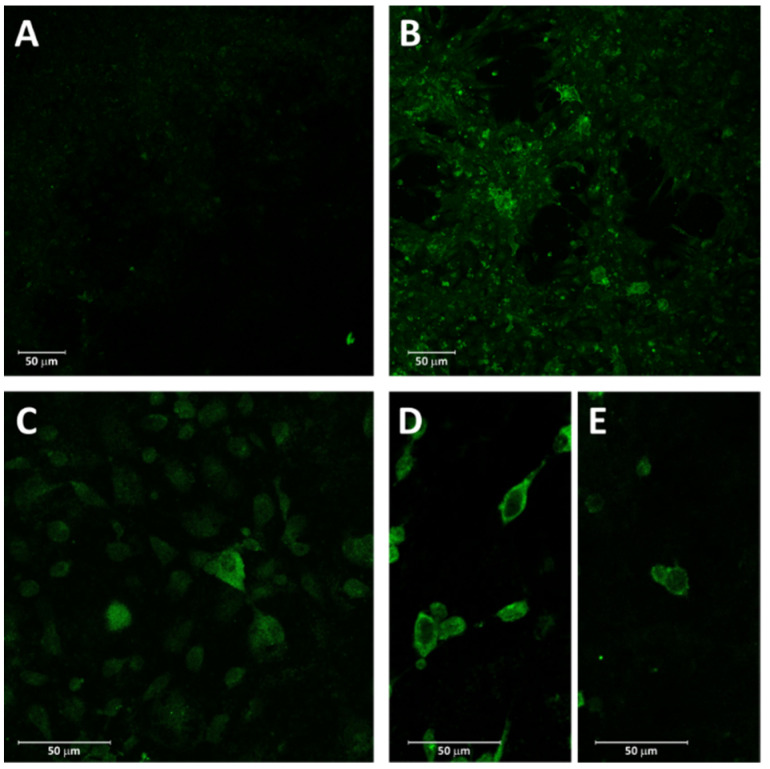
Plasmid transfection assay on EPC cells. (**A**) Negative control. (**B**) VHSV immunofluorescence signal at 5 days post-inoculation of EPC cells with VHS plasmid, 10×. (**C**) IHNV immunofluorescence signal at 7 days post-inoculation of EPC cells with IHN plasmid, 20×. (**D**,**E**) VHSV and IHNV signal at 5 days post-inoculation of EPC cells with VHS and IHN plasmids, 20×.

**Figure 2 vaccines-10-02062-f002:**
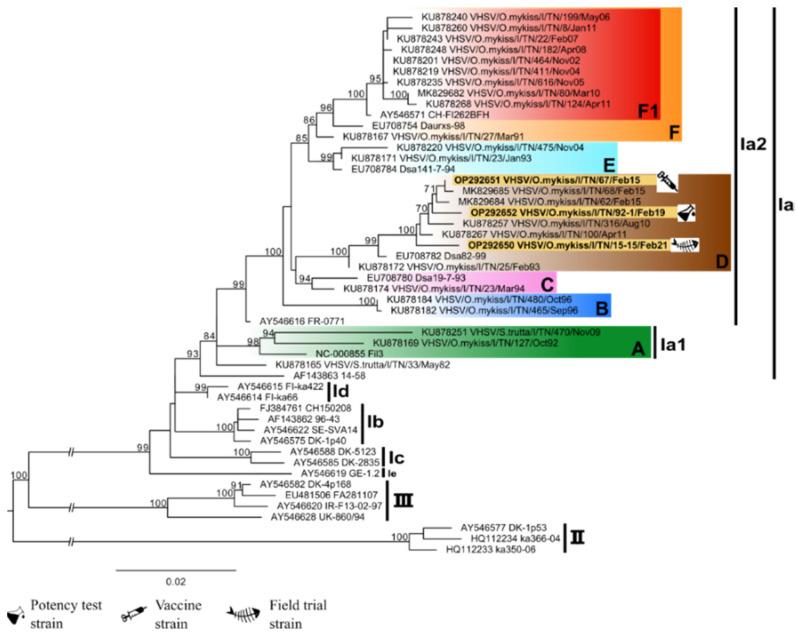
Phylogenetic tree based on the complete G gene sequence of VHSV. The strain used for the plasmid production, the strain used for the potency test, and the strain isolated during the field trial are highlighted in yellow. Vertical lines indicate the division into different genogroups and sub-lineages. The colored boxes highlight the different genetic clusters identified within sub-lineages Ia1 and Ia2 [19,26]. The posterior probability values ≥ 70 are reported for each node. The scale bar indicates the number of substitutions per site.

**Figure 3 vaccines-10-02062-f003:**
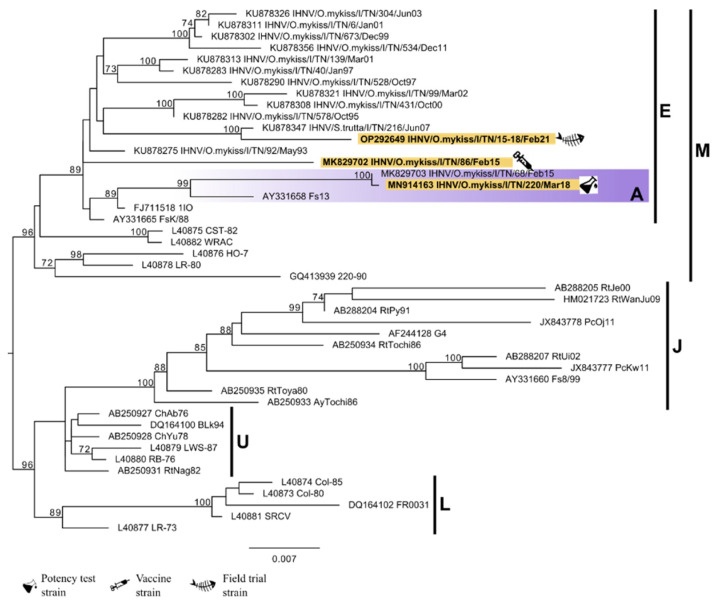
Phylogenetic tree based on the complete G gene sequence of IHNV. The strain used for the plasmid production, the strain used for the potency test, and the strain isolated during the field trial are highlighted in yellow. Vertical lines indicate the division into different genogroups and sub-lineages. The colored box highlights the genetic cluster identified within sub-lineage E [19,27]. Posterior probability values ≥ 70 are shown for each node. The scale bar indicates the number of substitutions per site.

**Figure 4 vaccines-10-02062-f004:**
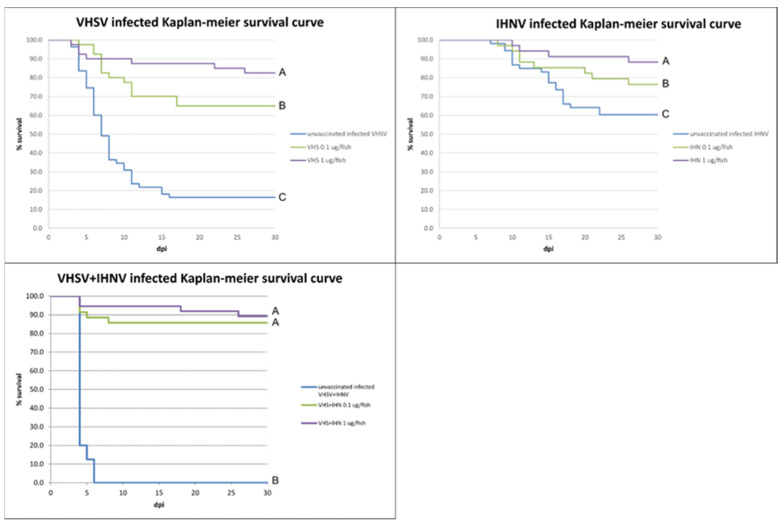
Controlled conditions potency test results. Kaplan–Meier survival curve for each challenged group, comparing mock vaccinated group with vaccinated groups. Different letters indicate statistical differences in pair-wise comparison (*p* < 0.01).

**Figure 5 vaccines-10-02062-f005:**
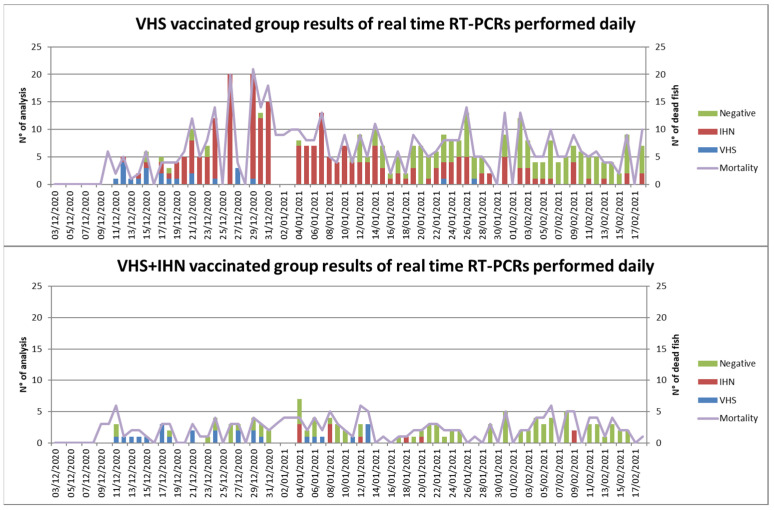
Daily results of the real-time RT-PCR performed on each sampled head kidney from VHS vaccinated group (**upper**) and VHS+IHN vaccinated group (**lower**). Positive results for VHSV are shown in blue, positive for IHNV in red, and negatives are represented in green. The purple line represents the daily mortality.

**Figure 6 vaccines-10-02062-f006:**
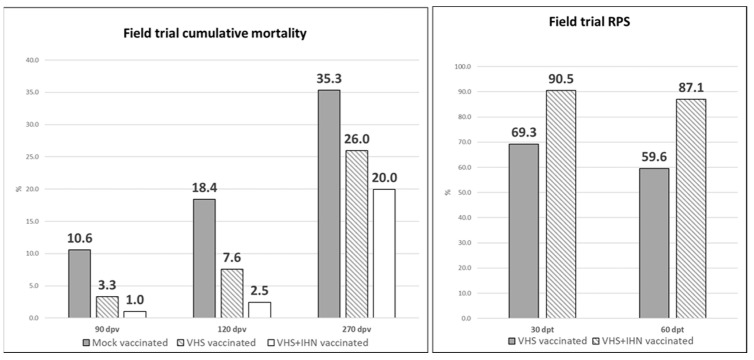
Field trial cumulative mortality and RPS at 90 and 120 dpv and cumulative mortality observed at 270 dpv. At each time, all the mortality values are statistically different (*p* < 0.001).

**Table 1 vaccines-10-02062-t001:** Viral strains and GenBank accession numbers for G gene sequences.

Use within the Study	Viral Strain	Year of Isolation	Host	Accession Number
VHS vaccine strain	VHSV/O.mykiss /I/TN/67/Feb15	2015	*O. mykiss*	OP292651
VHS potency strain	VHSV/O.mykiss/I/TN/92-1/Feb19	2019	*O. mykiss*	OP292652
VHS field trial strain	VHSV/O.mykiss/I/TN/15-15/Feb21	2021	*O. mykiss*	OP292650
IHN vaccine strain	IHNV/O.mykiss/I/TN/86/Feb15	2015	*O. mykiss*	MK829702
IHN potency strain	IHNV/O. mykiss/I/TN/220-2/Mar18	2018	*O. mykiss*	MN914163
IHN field trial strain	IHNV/O.mykiss/I/TN/15-18/Feb21	2021	*O. mykiss*	OP292649

**Table 2 vaccines-10-02062-t002:** Controlled condition experimental groups.

Tank	Group	Treatment	Challenge
1	Safety VHS	2 μg/fish VHS plasmid	Unchallenged
2	Safety IHN	2 μg/fish IHN plasmid	Unchallenged
3	Negative control	Unvaccinated	Unchallenged
4	VHS/IHN mock	Mock vaccinated	VHSV + IHNV bath challenged
5	VHS mock	Mock vaccinated	VHSV bath challenged
6	IHN mock	Mock vaccinated	IHNV bath challenged
7	VHS 0.1	0.1 μg/fish VHS plasmid	VHSV bath challenged
8	VHS 1	1 μg/fish VHS plasmid	VHSV bath challenged
9	IHN 0.1	0.1 μg/fish IHN plasmid	IHNV bath challenged
10	IHN 1	1 μg/fish IHN plasmid	IHNV bath challenged
11	VHS/IHN 0.1	0.1 μg/fish VHS plasmid + 0.1 μg/fish IHN plasmid	VHSV + IHNV bath challenged
12	VHS/IHN 1	1 μg/fish VHS plasmid + 1 μg/fish IHN plasmid	VHSV + IHNV bath challenged

**Table 3 vaccines-10-02062-t003:** PCR conditions and primers for plasmid and endogenous gene analysis.

PCR	Primer	Sequence 5’→3’	Amplicon Size	T Annealing (°C)
pVax1-vhsG	F1_(pVAX1)_VHSV R1_pVAX1_(VHSV)	CAATGGGAGTTTGTTTTGGCACCAA TGCAGTTGGAGGGATGAGTGTATAG	370	62
pVax1-ihnG	F1_(pVAX1)_IHNV R1_pVAX1_(IHNV)	GGCTAACTAGAGAACCCACTGCTTA GGATAGGCAATTAGTCCCCTGTTCT	327	58
Endogenous gene	B-Glob_FOR B-Glob_REV	TCCATGGACTCAGAGACACTTC GACACAACGACAGCCAGGAA	443	57

**Table 4 vaccines-10-02062-t004:** Controlled conditions trial: summary of the challenge dose, cumulative mortality, and relative percent survival (RPS) per experimental group. RPS = relative percent survival.

Group ID	Tank	Treatment	Challenge Virus and Dose	Mortality (%)	RPS
Safety test VHS	1	2 μg VHS plasmid	-	0	-
Safety test IHN	2	2 μg IHN plasmid	-	0	-
Negative control	3	100 μL PBS	-	0	-
Positive control VHS IHN	4	100 μL PBS	10^6.30^ TCID_50_/mL × (VHSV 92-1 + IHNV 220-2)	100	-
Positive control VHS	5	100 μL PBS	10^4.55^ TCID_50_/mL × VHSV 92-1	83.6	-
Positive control IHN	6	100 μL PBS	10^4.55^ TCID_50_/mL × IHNV 220-2	40.4	-
VHS 0,1	7	0.1 μg VHS plasmid	10^4.55^ TCID_50_/mL × VHSV 92-1	42.5	49.2
VHS 1	8	1 μg VHS plasmid	10^4.55^ TCID_50_/mL × VHSV 92-1	17.9	78.5
IHN 0,1	9	0.1 μg IHN plasmid	10^4.30^ TCID_50_/mL × IHNV 220-2	23.5	44.6
IHN 1	10	1 μg IHN plasmid	10^4.55^/mL × IHNV 220-2	11.1	72.5
VHS IHN 0,1	11	0.1 μg/fish VHS plasmid + 0.1 μg/fish IHN plasmid	10^6.05^ TCID50/mL × (VHSV 92-1+ IHNV 220-2)	14.3	87.5
VHS IHN 1	12	1 μg/fish VHS plasmid + 1 μg/fish IHN plasmid	10^5.80^ TCID50/mL × (VHS 92-1+ IHN 220-2)	10.8	90

**Table 5 vaccines-10-02062-t005:** VHS and IHN real-time RT-PCR results from spleen and kidney sampled from dead fish retrieved from tank 11 and 12. When positive, cycle threshold values (Ct) are given in brackets.

Tank	ID	Organ	VHSV	IHNV
11	1	Spleen	Pos (26.98)	Pos (30.55)
		Kidney	Pos (20.48)	Pos (27.31)
	2	Spleen	Pos (27.74)	Pos (32.69)
		Kidney	Pos (22.53)	Pos (28.92)
	3	Spleen	Pos (24.89)	Pos (32.54)
		Kidney	Pos (19.71)	Pos (28.82)
	4	Spleen	Pos (30.58)	Pos (29.03)
		Kidney	Pos (30.14)	Pos (33.55)
	5	Spleen	Pos (22.39)	**Neg**
		Kidney	Pos (22.39)	**Neg**
12	1	Spleen	Pos (29.67)	**Neg**
		Kidney	Pos (24.64)	**Neg**
	2	Spleen	Pos (30.06)	**Neg**
		Kidney	Pos (20.07)	**Neg**
	3	Spleen	**Doubt (36.00)**	**Neg**
		Kidney	Pos (33.65)	**Neg**
	4	Spleen	**Neg**	**Neg**
		Kidney	**Neg**	**Neg**

**Table 6 vaccines-10-02062-t006:** Results of the diagnostic examinations performed weekly on pools of moribund/dead fish collected from the mock vaccinated group up to 134 dpv. Numbers in brackets indicate the positive cycle threshold (ct) of real-time RT-PCR. dpv = day post-vaccination. Samples marked with * were evaluated by viral isolation on cell culture.

Timepoint	VHSV	IHNV	Bacteriological Examinations
64 dpv	Pos (18.79 ct)	**Neg**	**Neg**
69 dpv	Pos (19 ct)	**Neg**	**Neg**
76 dpv	Pos (22.54 ct)	**Neg**	**Neg**
84 dpv	Pos (22.94 ct)	Neg	Positive to *A.papoffii, A.sobria, A. bestiarum*
90 dpv	Pos (23.00 ct)	Pos (18.50 ct)	**Neg**
98 dpv	Pos *	Pos *	Positive to *A. sobria and P. shigelloides*
105 dpv	Pos (21.35 ct)		Positive to *A. sobria*
111 dpv	Pos (27.24 ct)	Pos (26.33 ct)	**Neg**
118 dpv	**Neg**	Pos (28.2 ct)	Positive to *A. dhakensis*
125 dpv	**Neg**	Pos (20.96 ct)	Positive to *Y. Ruckeri* biotype II
134 dpv	Pos (32.33 ct)	**Neg**	**Neg**

**Table 7 vaccines-10-02062-t007:** Results of the diagnostic examinations performed monthly on pools of moribund/dead fish collected from all experimental groups between 182 and 280 dpv. Numbers in brackets indicate positive cycle threshold (ct) of real-time RT-PCR. dpv = day post-vaccination.

Timepoint	Group	Real-Time RT-PCR VHSV	Real-Time RT-PCR IHNV	Bacteriological Examinations
182 dpv	Mock	Pos (35.28 ct)	Pos (20.86 ct)	**Neg**
	VHS	Pos (35.14 ct)	Pos (19.75 ct)	**Neg**
	VHS+IHN	Pos (31.22 ct)	Pos (18.67 ct)	**Neg**
210 dpv	Mock	**Neg**	Pos (24.85 ct)	Positive to *Y. Ruckeri* biotype II
	VHS	Pos (26.30 ct)	**Neg**	Positive to *Y. Ruckeri* biotype II
	VHS+IHN	**Neg**	Pos (31.69 ct)	Positive to *Y. Ruckeri* biotype II
232 dpv	Mock	**Neg**	Pos (27.90 ct)	**Neg**
	VHS	**Neg**	Pos (33.86 ct)	**Neg**
	VHS+IHN	**Neg**	Pos (27.40 ct)	**Neg**
280 dpv	Mock	**Neg**	Pos (31.77 ct)	**Neg**
	VHS	**Neg**	Pos (34.66 ct)	**Neg**
	VHS+IHN	**Neg**	**Neg**	**Neg**

**Table 8 vaccines-10-02062-t008:** Persistence of plasmids in epaxial muscle of rainbow trout collected at different days post-vaccination (dpv) during the field trial.

Trial		Plasmid Detection	pVax1-vhsG-Positive	pVax1-ihnG-Positive
	Time Point (dpv)			
**Potency test**	90		5 /5	5/5
**Field trial**	120		1/5	1/5
	160		3/5	3/5
	180		3/5	2/5
	210		2/5	2/5
	230		3/5	3/5
	260		4/5	0/5
	280		0/5	0/5
	320		6/15	6/15

## Data Availability

Data are available upon reasonable request to the authors.

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
