# Peer review of "Efficacy of DNA Vaccines in Protecting Rainbow Trout against VHS and IHN under Intensive Farming Conditions"

_vaccines, 2022, doi:10.3390/vaccines10122062_

Round 1
Reviewer 1 Report
The manuscript evaluated the efficacy of two DNA vaccines both in lab and field conditions, which was safe and had lower mortality. Though the study is quite interesting, there are serious flaws in the study design, data representation and analysis.
Several important points are written below.
1. Lines 111-113: "250 l", "50 l", please check.
2. Lines 118-119: What is the temperature at which cells are usually cultured? The authors did not include temperature in the cell culture conditions, but was maintained at 25℃ for transfection and 15℃ for immunofluorescence test. Please write clearly.
3. Lines 148-150: three experimental groups were control, 1 μg/fish of VHS plasmid, and 1 μg/fish of both VHS and IHN plasmids. Why is there no 1 μg/fish of IHN plasmid? This is not a complete trial design. Will the use of both plasmids negatively affect the IHNV plasmids? How can the authors prove this question without this group of trials?
4. In section 2.4: Why is there no group of "Safety VHS/IHN"? The same question, will the use of both plasmids negatively affect the IHNV plasmids? The results of this dat are necessary to evaluate the safety of the two plasmids combined.
5. In section 3.1, the plasmids of VHS were first detected at day 5, and from day 7 for IHN. In S2 (D-E), the combination of both were detected at days 5, why can the plasmids of IHN be detected in advance? Can two plasmids be transfected together to achieve this result? Please explain.
6. In section 3.1 and 2.4, in-house produced rabbit anti VHS and anti IHN sera were used to detect the expression of the G glycoprotein on EPC cells. But how did the author prepare the serum and whether the expression of G protein could be detected? And the WB result is also required.
7. In Table 4, the mortality of IHN challenge was only 40.4%, which was 83.6% of VHS challenge. If the virulence of the virus is too low, after the challenge test, some fish may have been infected with the virus but did not die, which will cause the calculated RPS value to be very different from the real result. Why not use a higher titer of virus? Another problem was that the titers of the virus were not consistent among these groups, how do you compare the results of different treatments?
8. In section 3.6, the author examined the metabolism of plasmid after immunization, how about the level of specific antibodies? The vaccines induce immune system of fish to produce antibodies which play an important role in fighting viral infections. Whether the combination of the two vaccines can induce higher levels of antibodies? How long do specific antibodies last in field trial?
Author Response
Authors thank the reviewer for the valuable and helpful comments and remarks.
Please see the attachment for the replies

Reviewer 2 Report
In the paper “Efficacy of DNA vaccines in protecting rainbow trout from VHS and IHN: experimental and field evidences”, the authors provide some valuable research data for DNA vaccines against VHS and IHN disease in rainbow trout at field trial.
The data is of value, nonetheless, some major issues arise:
1. Delete Template Description, such as “0. How to Use This Template”
2. Line 43: “VHS and IHN are classified as category C, D and E diseases”. Why the two diseases are classified into three categories? It is confusing. Rephrase it.
3. Line 65: put references [6], [7] together. This should be checked throughout the draft.
4. Put the IF detection of VHS and IHN plasmids expressed in EPC in main text along with section 3.1 instead of as Supplementary materials.
5. Section name “VHS and IHN plasmids” should be more specific, such as “VHS and IHN plasmids expression in EPC” or “VHS and IHN plasmids detection in EPC”.
6. In section 3.2. the name of virus strains used in this study should be listed in text except in the picture. Figure 1 and 2 can be combined into one figure as Figure 1A and 1B.
7. Section name 3.3 -3.7 should be more specific.
8. Fig 3 is too small to see the word in the figure.
9. Fig 4 should add a vertical coordinate on the right that refers to the mortality. Additionally, in Fig 4 the change comparison before and after vaccination cannot be visualized. Please make the picture clearer, or replace the more suitable chart form.
10. in Fig 5, add Error Bars to the Chart.
11. section 3.7 “Statistical Analysis” is a title of the method description, change it. This part should add some necessary charts to make it easy to understand.
12. line 581, what is the mean “group and 3 that…”
13. in discussion, this part should be divided into more paragraphs to make it easier to read
Author Response
Authors thank the reviewer for the valuable and helpful comments and remarks.
The Authors want to thank the referee for the valuable and helpful comments. All remarks have been addressed as described in the following.
- Delete Template Description, such as “0. How to Use This Template”
Done
- Line 43: “VHS and IHN are classified as category C, D and E diseases”. Why the two diseases are classified into three categories? It is confusing. Rephrase it.
VHS and IHN are classified as category C diseases as well as D and E, according to the EU CIR 2018/1882 and are subjected to the different prescriptions applicable to these different categories ( namely rules regarding Eradication programme on voluntary base (cat C), Movements (cat D) and Surveillance (E)). Thus, the sentence has not been modified.
- Line 65: put references [6], [7] together. This should be checked throughout the draft.
The manuscript has been checked and modified accordingly.
- Put the IF detection of VHS and IHN plasmids expressed in EPC in main text along with section 3.1 instead of as Supplementary materials.
Done, IF detection is now present in the manuscript as Figure 1
- Section name “VHS and IHN plasmids” should be more specific, such as “VHS and IHN plasmids expression in EPC” or “VHS and IHN plasmids detection in EPC”.
Done
- In section 3.2. the name of virus strains used in this study should be listed in text except in the picture. Figure 1 and 2 can be combined into one figure as Figure 1A and 1B.
Name of relevant viral strains are already reported in section 2.3 and Table 1 as well as their role within the study and their accession number. To avoid redundancy, the section 3.2 has been synthetized and we would like to maintain as It is. Regarding the pictures, due to long legends and dimension of the phylogenetic tree, again we would like to maintain them separated.
- Section name 3.3 -3.7 should be more specific.
Section names have been attributed according to the results detailed in each. Consequently, we don’t understand how to be more specific.
- Fig 3 is too small to see the word in the figure.
Fig 3 (now Fig 4) has been modified accordingly
- Fig 4 should add a vertical coordinate on the right that refers to the mortality. Additionally, in Fig 4 the change comparison before and after vaccination cannot be visualized. Please make the picture clearer, or replace the more suitable chart form.
A vertical coordinate bar referring to the mortality has been added to the graphs. Regarding the second remark, both graphs are reporting results or RT-PCR for VHSV/IHNV performed daily on dead or moribund fish in the two vaccinated groups after the start of the mortality. Thus no comparison between before and after the vaccination is intended to be displayed.
- in Fig 5, add Error Bars to the Chart.
In Fig 5 ( now 6) Error bars cannot be added or calculated since the standard error is not applicable ( cumulative mortality data at each time point of each group has not been calculated as a mean or as a result of any other calculation other than the sum of the daily mortality in each group).
- section 3.7 “Statistical Analysis” is a title of the method description, change it. This part should add some necessary charts to make it easy to understand.
Section 3.7 “Statistical Analysis” has been removed and the content has been moved to the section 3.5 “Field trial” in order to uniform it to what has been done for the statistical analysis of the controlled condition trial ( section 3.3 “controlled conditions trial”). Analysis were performed through the STATA 12 software and the output would have been displayed as Charts or Tables but they would make the manuscript longer without adding any useful detail.
- line 581, what is the mean “group and 3 that…”
This was a typo which has been corrected
- in discussion, this part should be divided into more paragraphs to make it easier to read
In order to revise the English writing, Discussion section has undergone an extensive rearrangement and shortened. Thus, we think that this remark is not anymore applicable.

Round 2
Reviewer 2 Report
I have only one suggestion about this version
in Fig 1. the scale bar representing the length should be added.
Author Response
Authors thank the reviewer for the suggestion which has been implemented accordingly.